# Outcomes of mechanically ventilated patients with COVID-19 associated respiratory failure

**Christopher S. King** [1]*, **Dhwani Sahjwani**[2], **A. Whitney Brown**[1], **Saad Feroz**[2], **Paula Cameron**[3], **Erik Osborn** [4], **Mehul Desai**[4], **Svetolik Djurkovic**[4], **Aditya Kasarabada**[4], **Rachel Hinerman**[4], **James Lantry**[4], **Oksana A. Shlobin**[1], **Kareem Ahmad**[1], **Vikramjit Khangoora**[1], **Shambhu Aryal**[1], **A. Claire Collins** [5], **Alan Speir**[6], **Steven Nathan**[1]

**1** Advanced Lung Disease and Transplant Program, Inova Fairfax Hospital, Falls Church, VA, United States of America, **2** Department of Pediatrics, Inova Fairfax Hospital, Falls Church, VA, United States of America, **3** Respiratory Therapy Department, Inova Fairfax Hospital, Falls Church, VA, United States of America, **4** Medical Critical Care Service, Inova Fairfax Hospital, Falls Church, VA, United States of America, **5** Advanced Lung Disease Research, Inova Fairfax Hospital, Falls Church, VA, United States of America, **6** Department of Cardiothoracic Surgery, Inova Fairfax Hospital, Falls Church, VA, United States of America

* Christopher.king@inova.org

## Abstract

### Purpose

The outcomes of patients requiring invasive mechanical ventilation for COVID-19 remain poorly defined. We sought to determine clinical characteristics and outcomes of patients with COVID-19 managed with invasive mechanical ventilation in an appropriately resourced US health care system.

### Methods

Outcomes of COVID-19 infected patients requiring mechanical ventilation treated within the Inova Health System between March 5, 2020 and April 26, 2020 were evaluated through an electronic medical record review.

### Results

1023 COVID-19 positive patients were admitted to the Inova Health System during the study period. Of these, 164 (16.0%) were managed with invasive mechanical ventilation. All patients were followed to definitive disposition. 70/164 patients (42.7%) had died and 94/164 (57.3%) were still alive. Deceased patients were older (median age of 66 vs. 55, p <0.0001) and had a higher initial d-dimer (2.22 vs. 1.31, p = 0.005) and peak ferritin levels (2998 vs. 2077, p = 0.016) compared to survivors. 84.3% of patients over 70 years old died in the hospital. Conversely, 67.4% of patients age 70 or younger survived to hospital discharge. Younger age, non-Caucasian race and treatment at a tertiary care center were all associated with survivor status.

### Conclusion

Mortality of patients with COVID-19 requiring invasive mechanical ventilation is high, with particularly daunting mortality seen in patients of advanced age, even in a well-resourced

**Data Availability Statement:** All relevant data are within the paper and its Supporting Information files.

**Funding:** The authors received no specific funding for this work.

**Competing interests:** The authors have declared that no competing interests exist.

health care system. A substantial proportion of patients requiring invasive mechanical ventilation were not of advanced age, and this group had a reasonable chance for recovery.

## Introduction

Since its start in late 2019 in Wuhan, China, the Coronavirus Disease 2019 (COVID-19) has blossomed into a worldwide pandemic, infecting 3 million people and killing over 200,000 [1]. The rapid spread of the disease has been paralleled by an explosion of publications on the topic, with over 7,500 publications produced by a PubMed search of "COVID-19" at the start of May 2020. Despite the intense interest and effort of the medical community to better understand and treat COVID-19, substantial gaps in our knowledge of the disease remain. One particularly important area lacking clarity is the prognosis of COVID-19 patients with acute respiratory failure requiring invasive mechanical ventilation (IMV). Mortality estimates vary substantially, ranging from 16 to 97%, with multiple studies citing mortality in excess of 50% [2–7].

These reports have led to alarming headlines in the lay press, such as "Most COVID Patients Placed On Ventilators Died, New York Study Shows," the title of an article recently published in U.S. News and World Report [8]. However, there are significant limitations to the available literature. Much of it is derived from centers outside of the United States, where the standard of care and patient populations may differ from those seen in most United States hospitals. In addition, many of these reports come from hospitals that were experiencing a major surge in patient volumes and were forced to use suboptimal equipment and staffing models that varied considerably from typical practice. Using the data from these studies to provide counseling to families and patients with impending respiratory failure may provide an unrealistically grim estimate of the chance of survival, leading some to forego potentially life-saving treatment. We sought to delineate the survival of patients with acute respiratory failure from COVID-19 requiring IMV in a United States hospital system with a high volume of COVID-19 patients, but not surging to a capacity that outstripped the ability to provide critical care in line with the conventional standard of care.

## Methods

Data on all COVID-19 positive patients who were placed on IMV for acute respiratory failure within the Inova Health System in Northern Virginia between March 5, 2020 and April 26, 2020 was collected. Outcomes were reassessed on August 19, 2020. The Inova Health System consists of five hospitals including a large tertiary care center and four community hospitals. COVID-19 infection was confirmed by a positive result on polymerase chain reaction testing from either a nasopharyngeal or lower respiratory tract sample. There were no transfers of COVID-19 patients into or out of the Inova Health System during the study period. Transfers within the health system to the tertiary care hospital were analyzed as a single hospitalization attributed to the accepting facility.

All data was collected from the electronic medical record (Epic®). Data collected included patient demographics (race, ethnicity age, gender), comorbidities, adjunctive respiratory treatments [paralysis, prone positioning, inhaled pulmonary vasodilators including inhaled nitric oxide, inhaled epoprostenol, and extracorporeal membrane oxygenation (ECMO)], COVID-19 targeted treatments (clinical trial enrollment, use of toculizumab, hydroxychloroquine, remdesivir, convalescent plasma), secondary infections, and outcomes [extubation, ventilator

days, discharge, death, hospital length of stay, development of acute kidney injury, and need for renal replacement therapy (RRT)]. Cause of death was determined by chart review. Immunosuppressed individuals were comprised of solid organ transplant recipients, patients on active chemotherapy, and individuals on chronic immunosuppression for any other indication (at an equivalent of prednisone 20 mg daily or higher). In the event of reintubation, ventilator length of stay was calculated from date of the initial intubation until final extubation. Outcomes were unknown for a subset of patients who remained on ventilator support or hospitalized at the time of data censoring.

The initial and highest values of laboratory data including white blood cell count, ferritin, C-reactive protein (CRP), and D-dimer were also collected. Values listed as greater than or less than the maximal or minimal test value were listed as that cutoff value (e.g. d-dimer > 20 was recorded as 20). Finally, respiratory/ventilator parameters including highest positive end expiratory pressure (PEEP), highest fraction of inspired oxygen (FiO2) required and lowest ratio of pulmonary arterial oxygen tension to FiO2 (P/F ratio) were collected.

The strategy for management of acute respiratory failure was fairly homogenous across the system. Efforts were made to avoid intubation where feasible with the use of reservoir cannulas and high flow nasal cannula (HFNC). Non-invasive ventilation was largely avoided early on due to concerns about aerosolization of the virus, but was increasingly utilized over time. Inhaled nitric oxide was delivered via HFNC in a number of patients. Self-proning was incorporated where appropriate in non-intubated patients. If these measures failed and intubation was required, patients were typically managed initially with moderate PEEP (10–12 cm $H_2O$) and a lung protective ventilator strategy targeting tidal volumes of 6 mL/Kg of ideal body weight (IBW) and plateau pressures < 30 cm $H_2O$. In patients with compliant lungs, tidal volumes were often liberalized to 7–8 mL/Kg IBW as long as plateau pressure remained < 30 cm $H_2O$. Alternatively, some patients were switched to pressure control ventilation. Ultimately the ventilator strategy was left to the discretion of the attending intensivist. Paralysis was frequently utilized in patients with severe ARDS, defined as P/F ratio < 150. Prone positioning was also utilized in patients with a P/F ratio < 150 who required FiO2 of ≥ 60% and PEEP ≥ 10 cm $H_2O$. Patients were maintained in the prone position for 16 hours or longer when performed. A conservative fluid strategy was utilized whenever possible, but was not undertaken at the expense of worsening shock. Use of inhaled pulmonary vasodilator therapy was poorly standardized and left to the discretion of the attending intensivist. The choice of sedation and analgesia was also implemented at the discretion of the attending intensivist and was targeted to a Richmond Agitation Sedation Scale (RASS) of 0 to -2 [9]. Patients were considered for venovenous (VV) ECMO if age < 60 years old, on IMV < 10 days, had a P/F ratio < 100 and/or failed lung protective ventilation, despite neuromuscular blockade and prone positioning, or had recalcitrant hypercapnic acidosis affecting perfusion.

Treatments targeting COVID-19 disease were administered at the discretion of the attending physician and were subject to availability. Treatments utilized included hydroxychloroquine, toculizumab, convalescent plasma, remdesivir (either compassionate use or via clinical trial), and sarilumab via clinical trial. Need for and duration of antimicrobial agents was dictated by the attending intensivist, often with input from an Infectious Disease specialist. Use of corticosteroids and anticoagulants was poorly standardized.

This study was approved by the institutional review board (Inova Health System IRB # U20-05-4061). Continuous and categorical variables were presented as the median (IQR) and n (%), respectively, with the exception of length of stay data which was presented as the mean value. The Mann-Whitney U test, Chi-squared, or Fischer's exact test were used to compare differences between survivors and non-survivors where appropriate. A p value < 0.05 was considered statistically significant. Univariate and multivariate logistic regression analysis of

factors potentially associated with mortality were performed. Variables were dropped from the model through use of the likelihood ratio test. All statistical analyses were performed using STATA version 12 (StataCorp LP; College Station, TX, USA).

## Results

A total of 1023 COVID-19 positive patients were admitted in our health system during the study period. Exact numbers of patients admitted to ICU beds could not be discerned, as our health system adapted to a contingency status where critically ill patients were managed in both ICU and intermediate care beds. A total of 164 COVID-19 positive patients in our health system required invasive mechanical ventilation during the study period, representing 16.0% of admitted COVID-19 patients. Ninety-four (57.3%) of patients survived to hospital discharge. Table 1 describes the baseline demographics of the IMV patients. The most notable statistically significant demographic difference between the deceased and survivor groups (defined as alive at the time of data censoring) was age, with median ages of 67 vs. 55, respectively. Table 1 provides laboratory and ventilator data on the cohort. The only observed laboratory differences between deceased and survivor groups were a higher initial d-dimer (2.22 vs. 1.31, p = 0.005) and peak ferritin levels (2998 vs. 2077, p = 0.016). No significant difference was found in peak d-dimer, initial ferritin, or initial or peak CRP and WBC. The entire cohort had severe hypoxemic failure with 51.8% having a PaO2/Fio2 ratio < 100 and 86% with a PaO2/FiO2 ratio <200. The deceased cohort had a lower nadir PaO2/FiO2 ratio at 85.5 compared to 115.6 (p = 0.019) for the survivor cohort.

The average time from admission to intubation was 2.5 days (± 3.0 SD) (Range: 0–18 days); however, 43 patients (26%) were intubated on the day of admission. There was no significant difference in the mean time to intubation between the deceased patients and survivors (2.4 vs. 2.7 days, p = 0.54). The mean duration of ventilator support for survivors was 14.6 days (± 12 SD) (Range: 1–59 days). The mean length of stay for patients discharged alive was 24.5 days (± 14.8 SD) (Range: 7–86 days). The ventilator and hospital LOS for deceased patients were 9.3 (± 6.95 SD) and 11 (± 10.04 SD) days respectively. For those who died, the cause of death was hypoxemic respiratory failure in the majority of patients (n = 56, 80%). Other causes of death included shock (n = 9, 12.8%), cerebrovascular accident (n = 2, 2.8%), bowel ischemia (n = 1, 1.4%), subarachnoid hemorrhage (n = 1, 1.4%), and complications of ECMO cannulation (n = 1, 1.4%).

A total of 16 patients in this cohort were treated with both ECMO and IMV, representing 9.7% of the total cohort. 81.25% of the patients on ECMO survived, as compared to 54.7% of those managed with IMV alone. Table 2 summarizes the outcomes for included patients.

Table 3 displays the age distribution for deceased patients and survivors. Patients over 70 accounted for over one third of the deaths in the cohort. In fact, over 80% of patients over age 70 died. In the multivariate analysis, the odds ratio of death was 1.07 for age meaning that for every one point increase in age, there was a seven percent increase in the odds of death. No differences were seen with regard to gender, assessed comorbidities, or BMI. White race was found to be associated with deceased status when compared to other races. A substantial portion of the overall cohort reported Hispanic ethnicity (36%); Hispanic ethnicity appeared to be associated with high likelihood of survivor status in comparison to non-Hispanic ethnicity, although that association did not remain after multivariate analysis, likely due to the relationship between race and ethnicity.

Fifty-two (55.3%) of the survivors were managed at a tertiary care center, while only 27 (38.6%) deceased patients were cared for at the same tertiary care center. Patients managed at a tertiary care center were statistically more likely to be survivors (p = 0.034). Patients managed

**Table 1. Baseline demographics of COVID-19 patients managed with invasive mechanical ventilation stratified by survivor status.**

| | Total (N = 164) | Non-Survivors (N = 70) | Survivors (N = 94) | p value* |
|---|---|---|---|---|
| Male Gender (%) | 107 (65%) | 43 (40%) | 64 (60%) | 0.38 |
| Age, median(IQR) | 58 (18), | 67 (22), | 55 (14), | **<0.0001** |
| | Range: 16–91 | Range: 29–91 | Range: 16–84 | |
| Race (N = 161) | White 57 (35.4%) | White 36 (51.4%) | White 21(23.1%) | **<0.0001** |
| | Black 42 (26.1%) | Black 21 (30.0%) | Black 21 (23.1%) | |
| | Asian 23 (14.3%) | Asian 8 (11.4%) | Asian 15 (16.5%) | |
| | Other 39 (24.2%) | Other 5 (7.1%) | Other 34 (37.4%) | |
| Hispanic Ethnicity | 59 (36%) | 18 (26%) | 41 (43%) | **0.018** |
| BMI, median (IQR) | 30 (10.4) | 30 (11.8) | 30 (9.5) | 0.8315 |
| | Range: 13–56 | Range: 13–56 | Range: 18–53 | |
| Comorbidities (%) | | | | |
| HTN | 85 (52%) | 40 (57.1%) | 45 (47.9%) | 0.24 |
| HLD | 47 (28.7%) | 21 (30.0%) | 26 (27.7%) | 0.74 |
| DM | 56 (34.1%) | 24 (34.3%) | 32 (34.0%) | 0.97 |
| CAD | 11 (6.7%) | 7 (10.0%) | 4 (4.3%) | 0.15 |
| ESRD | 5 (3.1%) | 4 (5.7%) | 1 (1.1%) | 0.09 |
| Immunosuppressed | (%) | (%) | (%) | |
| Obesity (BMI ≥ 30) | (%) | (%) | (%) | |
| Morbid Obesity (BMI ≥ 35) | (%) | (%) | (%) | |
| WBC (X $10^9$/L) | | | | |
| Initial | 7.44 (3.81) | 7.87 (4.73) | 7.33 (3.48) | 0.49 |
| Peak | 16.36 (8.24) | 17.9 (12.14) | 15.6 (6.77) | 0.07 |
| CRP (mg/dL) | | | | |
| Initial | 14.1 (13.8) | 14 (14.65) | 14.1 (14.1) | 0.57 |
| Peak | 27.3 (14.6) | 26.4 (16.7) | 28.2 (12.5) | 0.28 |
| D-Dimer (ng/mL) | | | | |
| Initial | 1.51 (2.35) | 2.22 (3.8) | 1.31 (1.7) | **0.0053** |
| Peak | 6.82 (10.58) | 7.11 (11.9) | 5.96 (10.2) | 0.22 |
| Ferritin (ng/mL) | | | | |
| Initial | 1106 (1913) | 1141 (1920) | 1033 (1797) | 0.86 |
| Peak | 2456 (3904) | 2998 (8145) | 2077 (3008) | **0.0158** |
| Lowest PaO2/FiO2ratio | 98.3 (89.1) | 85.5 (67.5) | 115.6 (84.6) | ***0.019*** |
| ≤ 100 | n = 85,51.8% | n = 45,64.2% | n = 40,42.6% | |
| 101–200 | n = 56,34.1% | n = 16,22., % | n = 40,42.6% | |
| 201–300 | n = 13,7.9% | n = 3, 4.3% | n = 10,10.6% | |
| >300 | n = 10,6.1% | n = 6,8.6% | n = 4,4.3% | |
| Maximum PEEP | 12 (5) | 13 (6) | 12 (4) | 0.26 |
| | Range: 5–28 | Range: 5–24 | Range: 5–28 | |

Abbreviations: BMI = Body mass index; IQR = Interquartile Range; HTN = Hypertension; HLD = Hyperlipidemia; DM = Diabetes mellitus; CAD = Coronary Artery Disease; CKD = Chronic kidney Disease WBC = White blood cell count, CRP = C-reactive protein, PEEP = Positive end-expiratory pressure.

* Data shown as median (IQR) or percentage (%).

Comparisons between survivors and non-survivors based on Wilcoxon Rank-sum (Mann-Whitney) test for continuous variables and Chi-square test for categorical variables.

**Table 2. Patient outcomes.**

|  | Deceased | Survivor* |
|---|---|---|
| **Total Cohort (n = 164)** | 70 (42.7%) | 94 (57.3%) |
| **IMV only (n = 148)** | 67 (45.2%) | 81(54.7%) |
| **ECMO only (n = 16)** | 3 (18.8%) | 13 (81.3%) |

Abbreviations: ECMO = Extracorporeal membrane oxygenation; IMV = Invasive mechanical ventilation.

at the tertiary care center had access to both extracorporeal membrane oxygenation (ECMO) and clinical trial enrollment. Table 4 summarizes various treatments provided to the cohort of patients. Survivors were more likely to receive tocilizumab or to be on ECMO, although the survival advantage of ECMO use did not persist after adjustment for multiple variables in the logistic regression analysis as it was correlated with access to tertiary care. Table 5 summarizes the findings of the univariate and multivariate analysis. Treatment with tocilizumab was associated with improved survival with an adjusted odd's ratio of 0.42 (p = 0.45). There was a trend toward increased need for CRRT in deceased patients (35.7% versus 18.1%, p = 0.07). Nearly all patients (n = 152, 92.1%) received antimicrobials for some duration of time. Only 29 patients (17.6%) had confirmed, culture positive secondary infections, 11 (17.4%) in deceased patients and 18 (17.6%) in surviving patients. One patient developed *Clostridium difficle* infection.

## Discussion

The primary finding of our analysis is that mortality in COVID-19 patients requiring mechanical ventilation is high, particularly in patients of advanced age. Our study adds to the available literature on outcomes. Our data has the advantage of following all included patients to either death or hospital discharge, whereas most existing studies include patients which are still hospitalized.

Despite the high mortality, the outcomes of mechanically ventilated patients in our health system compare favorably to those reported elsewhere. For instance, in the report by Richardson, et. al. on the Northwell Health System in New York, 1151 patients required IMV. At the time of their report, 24.5% of the patients had died, while only 3.3% were discharged alive, and 72% remained in the hospital [2]. If only those with a confirmed endpoint (death or discharge) from this cohort are analyzed, the reported mortality rate for patients requiring IMV is 88.1% [2]. Data from Wuhan, China reported by Zhou and colleagues found that 31 out of 32 patients (96.8%) treated with IMV died [3]. ICNARC, the Intensive Care National Audit and Research Centre from the United Kingdom, reported that 56.8% of patients treated with "advanced respiratory support", which can include high flow oxygen, non-invasive ventilation, ECMO or IMV, died in the hospital [7]. Graselli, et. al. reported on 1591 patients hospitalized in the ICU

**Table 3. Age distribution of cohort stratified by survivor status.**

| Age | Total | Deceased | Survivors* |
|---|---|---|---|
| ≤ 40 | 14 (8.5%) | 3 (4.3%) | 11 (11.7%) |
| 41–50 | 31 (18.9%) | 7 (10%) | 24 (25.5%) |
| 51–60 | 49 (29.8%) | 18 (25.7%) | 31 (33.0%) |
| 61–70 | 38 (23.2%) | 15 (21.4%) | 23 (24.4%) |
| > 70 | 32 (19.5%) | 27 (38.6%) | 5 (5.3%) |

*Survivors were patients alive at the time of the analysis

**Table 4. Comparison of treatments received by deceased vs. survivor cohorts.**

| | Total | Deceased | Survivors* | p value |
|---|---|---|---|---|
| | (N = 164) | (N = 70) | (N = 94) | |
| Care at Tertiary Center | 79 (48.2%) | 27 (38.6%) | 52 (55.3%) | **0.034** |
| Antimicrobials | 152 (92.7%) | 66 (94.3%) | 86 (91.5%) | 0.50 |
| Hydroxychloroquine | 132 (80.5%) | 52 (74.3%) | 80 (85.1%) | 0.08 |
| Toculizumab | 49 (29.9%) | 13 (18.6%) | 36 (38.3%) | **0.006** |
| Clinical Trial | 15 (9.2%) | 4 (5.7%) | 11 (11.7%) | 0.19 |
| Inhaled Pulmonary Vasodilators | 23 (14.0%) | 4 (5.7%) | 19 (20.2%) | **0.008** |
| Paralysis | 61 (37.2%) | 25 (35.7%) | 36 (38.3%) | 0.74 |
| Prone Positioning | 100 (61.0%) | 44 (62.9%) | 56 (59.6%) | 0.67 |
| CRRT | 42 (25.6%) | 25 (35.7%) | 17 (18.1%) | **0.011** |
| ECMO | 16 (9.7%) | 3 (4.3%) | 13 (13.8%) | **0.042** |

Abbreviations: ECMO = Extracorporeal membrane oxygenation; CRRT = Continuous renal replacement therapy

* Comparisons between survivors and non-survivors based on Chi-square tests

in the Lombardy Region of Italy [5]. They do not specifically report mortality for those managed with IMV, although the majority (72%) required IMV. At the time of data reporting, their ICU mortality rate was 26%, although 58% of the patients were still in the ICU. Finally, a report from Seattle, Washington, USA included data on 24 critically ill COVID-19 patients, 18 of whom required IMV [4]. At the time of data censoring, 50% of the patients died and 5 of 18 (27.7%) were still on mechanical ventilation.

We feel it is particularly reassuring that the death rate in our cohort was not higher, given our system strategy of avoiding intubation unless patients were truly unable to maintain their oxygenation or ventilation despite aggressive management with non-invasive measures. We managed patients with self-proning, inhaled pulmonary vasodilators, and high flow oxygen to avoid intubation whenever possible. Given this approach, the cohort of patients managed with IMV was likely sicker than those reported in some of series and could be expected to have less

**Table 5. Odds ratios of death among mechanically ventilated COVID-19 patients.**

| | Odds Ratio | P value |
|---|---|---|
| | (95% Confidence Interval) | |
| **Unadjusted:** (N = 164) | | |
| Age | 1.07 (1.04–1.10) | **<0.0001** |
| Male Gender | 0.75 (0.39–1.43) | 0.38 |
| Non-Caucasian Race (N = 161) | 0.47 (0.34–0.64) | **<0.0001** |
| Hispanic Ethnicity | 0.45 (0.23–0.88) | **0.0170** |
| Tertiary Care Hospital | 0.51 (0.27–0.95) | **0.0332** |
| Enrollment in Clinical Trial | 0.46 (0.14–1.50) | 0.18 |
| ECMO | 0.28 (0.08–1.02) | **0.03** |
| Tocilizumab | 0.37 (0.18–0.76) | **0.0055** |
| Hydroxychloroquine | 0.51 (0.23–1.10) | 0.09 |
| ***Adjusted:** (N = 161) | | |
| Age | 1.06 (1.03–1.10) | **<0.0001** |
| Non-Caucasian Race | 0.54 (0.38–0.77) | **0.001** |
| Tertiary Care Hospital | 0.44 (0.20–0.94) | **0.034** |
| Tocilizumab | 0.42 (0.18–0.98) | **0.045** |

favorable outcomes. The extreme nature of the severity of illness in our cohort is supported by the median lowest PaO2/FiO2 ratio of 98.3 and the fact that > 85% of the patients had a PaO2/FiO2 ratio < 200.

Patients of advanced age account for the majority of deaths in our cohort. Of 32 patients age 70 of older, 84.3% were deceased at the time the data was censored. However, only 32.6% of patients < 70 years old and 22% of those < age 50 died in the hospital. Indeed, for every increasing year of age in our cohort, there was a 7% increase in the odds of death based on our multivariate logistic regression model. This relationship between advancing age and odds of mortality is consistent with other reports. There are likely multiple reasons for this including more co-morbidities, worse baseline functional status, and variations in the aggressiveness of goals of care. Notably, patients treated with tocilizumab were half as likely to die, although we suspect this result is secondary to inadequate variable control as the recent randomized, double-blinded, placebo controlled COVACTA trial failed to demonstrate a mortality benefit in COVID positive patients treated with tocilizumab [10]. A large number of patients within our cohort were offered adjunct treatments via clinical trials and/or ECMO. While the small number of patients within this cohort does not allow us to make assertions which of these treatment strategies provided the most benefit, the overall survival within our cohort likely supports the notion that availability of advanced interventions at a tertiary care center may play a significant role in improving patient outcomes.

Our study does have some limitations which should be acknowledged.

One limitation of our study is that it may not be generalizable to all health systems. Our health care system has a well-resourced, well-structured and dedicated medical critical care service with high volumes and experience treating ARDS, and hence adept at the application of best practices including proning and lung protective strategies. Additionally, we have a robust and experienced, high volume ECMO program at our tertiary care hospital which bolstered our outcomes. Also, our approach to treatment of hypoxemic respiratory failure prior to intubation may differ from those of other hospitals. Given our aggressive approach to use of noninvasive strategies including self-proning and HFNC, it is possible that our cohort of patients requiring IMV may in fact be sicker than those reported elsewhere. Another issue is that we were unable to provide specific data on use of HFNC or NIV prior to intubation. As mentioned in the methods, our center evolved to a strategy of delayed extubation and aggressive HFNC support relatively early in our surge, although our pre-intubation management strategy was somewhat in flux over time. Finally, since patients frequently had evolving physiology with variable lung compliance and ventilator settings, we are unable to provide specific details on these parameters. However, our ventilator management was in keeping with current best practices and therefore we believe that our survival estimates would be reproducible in similar health care systems.

In conclusion, the need for IMV in COVID-19 is associated with a high mortality in patients with COVID-19. However, successful outcomes are possible, with over 70% of patients younger than 70 still alive at the time of data censoring.

## Supporting information

**S1 Data.**
(XLSX)

## Acknowledgments

The study team would like to thank the physicians, nurses, respiratory therapists, pharmacists and ancillary care services who have tirelessly provide care for COVID-19 patients within the Inova health system.

## Author Contributions

**Conceptualization:** Christopher S. King, Paula Cameron, Erik Osborn, Mehul Desai, Svetolik Djurkovic, Aditya Kasarabada, Rachel Hinerman, James Lantry, Oksana A. Shlobin, Kareem Ahmad, Vikramjit Khangoora, Shambhu Aryal, Alan Speir, Steven Nathan.

**Data curation:** Christopher S. King, Dhwani Sahjwani, A. Whitney Brown, Saad Feroz, Paula Cameron.

**Formal analysis:** Christopher S. King, Steven Nathan.

**Investigation:** Christopher S. King.

**Methodology:** A. Whitney Brown, Saad Feroz, Erik Osborn, Mehul Desai, Svetolik Djurkovic, Aditya Kasarabada, Rachel Hinerman, James Lantry, Oksana A. Shlobin, Kareem Ahmad, Vikramjit Khangoora, Shambhu Aryal, A. Claire Collins, Alan Speir, Steven Nathan.

**Project administration:** Christopher S. King, A. Claire Collins.

**Supervision:** Christopher S. King.

**Validation:** Christopher S. King.

**Writing – original draft:** Christopher S. King, A. Whitney Brown, Steven Nathan.

**Writing – review & editing:** Christopher S. King, Dhwani Sahjwani, A. Whitney Brown, Saad Feroz, Paula Cameron, Erik Osborn, Mehul Desai, Svetolik Djurkovic, Aditya Kasarabada, Rachel Hinerman, James Lantry, Oksana A. Shlobin, Kareem Ahmad, Vikramjit Khangoora, Shambhu Aryal, A. Claire Collins, Alan Speir, Steven Nathan.

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
