## [Decision Letter · Decision Letter 0]

18 Aug 2020

PONE-D-20-21719

Outcomes of Mechanically Ventilated Patients with COVID-19 Associated Respiratory Failure

PLOS ONE

Dear Dr. King,

Thank you for submitting your manuscript to PLOS ONE. After careful consideration, we feel that it has merit but does not fully meet PLOS ONE’s publication criteria as it currently stands. Therefore, we invite you to submit a revised version of the manuscript that addresses the points raised during the review process.

We look forward to receiving your revised manuscript.

Kind regards,

Andrea Ballotta

Academic Editor

PLOS ONE

Additional Editor Comments:

Thanks for your submission. On the basis of the reviewer's comments i deem the paper not suitable for publication. It needs to be strongly revised. We don't need other partial data. WE need data about medium and long term outcome. The COVID 19 period has also been a sort of nightmare for science. WE must do much better.

2. In the ethics statement in the manuscript and in the online submission form, please provide additional information about the patient records used in your retrospective study.

Specifically, please ensure that you have discussed whether all data were fully anonymized before you accessed them and/or whether the IRB or ethics committee waived the requirement for informed consent.

If patients provided informed written consent to have data from their medical records used in research, please include this information.

3. For studies involving humans categorized by race/ethnicity, age, disease/disabilities, religion, sex/gender, sexual orientation, or other socially constructed groupings, authors should:

a) Explicitly describe their methods of categorizing human populations,

b) Define categories in as much detail as the study protocol allows,

c) Justify their choices of definitions and categories,

d) Explain whether (and if so, how) they controlled for confounding variables such as socioeconomic status, nutrition, environmental exposures, or similar factors in their analysis, and

e) Update outmoded terms and potentially stigmatizing labels to more current, acceptable terminology.

Examples: “Caucasian” should be changed to “white” or “of [Western] European descent” (as appropriate).

Reviewers' comments:

Reviewer's Responses to Questions

**Comments to the Author**

1. Is the manuscript technically sound, and do the data support the conclusions?

Reviewer #1: Partly

Reviewer #2: Yes

2. Has the statistical analysis been performed appropriately and rigorously? 

Reviewer #1: Yes

Reviewer #2: Yes

3. Have the authors made all data underlying the findings in their manuscript fully available?

Reviewer #1: Yes

Reviewer #2: Yes

4. Is the manuscript presented in an intelligible fashion and written in standard English?

Reviewer #1: Yes

Reviewer #2: Yes

5. Review Comments to the Author

Reviewer #1: Dear authors,

Thanks for your manuscript. I strongly belive in your message but unfortunately you referred to partial data, particularly about the patient's outcome. My suggestion is to obtain all patient's outcome, complications and ventilation management data and to revised the manuscript results.

Reviewer #2: Thanks for your paper.

To be precise in your analysis you might put the following details in the paper.

I wuold be important to indicate how much time patients has been treated before intubation ( sintoms, NIV and High flow cannula time).

Which was the cutoff for intubation, it was differnt over the time ?

When you indicate the ventilation it has been inappropriate only define the ammount of on ml/kg and plateu.

Is more convenient to indicate at least the driving pressure or better the compliance oder a partition with esophageal baloon.

Which were the indication for ECMo? Murray score Oxigenation index....? And which was the ECMO capability in the number of the admission in ICU .

Finally the big limitation you have also underline is the lack of long term survival which is important to evaluate the treatment.

6. PLOS authors have the option to publish the peer review history of their article (what does this mean?). If published, this will include your full peer review and any attached files.

Reviewer #1: **Yes: **Mirko Belliato

Reviewer #2: No

---

## [Author Response · Author response to Decision Letter 0]

8 Sep 2020

To the Editor,

Thanks to both you and your reviewers for your thoughtful and favorable review of our manuscript. We are pleased to provide a revised version for subsequent review. With this revision we will submit a de-identified version of our dataset as requested. We have attempted to answer the reviewers’ comments below. We look forward to your review. 

 Respectfully,

 Christopher King, MD

Reviewer #1: Dear authors,

Thanks for your manuscript. I strongly believe in your message but unfortunately you referred to partial data, particularly about the patient's outcome. My suggestion is to obtain all patient's outcome, complications and ventilation management data and to revised the manuscript results.

Thank you for your kind words with regards to our manuscript. Initially we hoped to get this manuscript out early to provide information to clinicians with regards to outcomes in intubated COVID-19 patients, but as you point out, a complete data set including all outcomes is more informative. We have now updated all patient outcomes so that all included patients have a clinical outcome of either death or discharge from the hospital. All statistics and tables have been updated in accordance with this change as well. To our knowledge this makes our manuscript the first to report definitive outcomes on an entire cohort of patients with COVID-19 requiring invasive mechanical ventilation. 

Reviewer #2: Thanks for your paper.

To be precise in your analysis you might put the following details in the paper.

I wuold be important to indicate how much time patients has been treated before intubation ( sintoms, NIV and High flow cannula time).

We include in the manuscript the time from admission to intubation for all included patients. This is provided in the results section where it states: 

“The average time from admission to intubation was 2.5 days (± 3.0 SD) (Range: 0-18 days); however, 43 patients (26%) were intubated on the day of admission. There was no significant difference in the mean time to intubation between the deceased patients and survivors (2.4 vs. 2.7 days, p = 0.54).” 

We are unable to provide data on time from symptom onset to intubation. Unfortunately, we are also limited in our ability to report specific data on use of HFNC and NIV. We do mention in the methods section that use of NIV was relatively uncommon. We have added a statement in the limitations section to address this. See the next comment for what was added.

Which was the cutoff for intubation, it was differnt over the time ?

Very early on we had a low threshold for intubation but quickly evolved to a strategy of delayed intubation and reliance on HFNC. The vast majority of patients would have been managed with a greater reliance on HFNC. We added the following to the limitations section “Another issue is that we were unable to provide specific data on use of HFNC or NIV prior to intubation. As mentioned in the methods, our center evolved to a strategy of delayed extubation and aggressive HFNC support relatively early in our surge, although our pre-intubation management strategy was somewhat in flux over time.” 

When you indicate the ventilation it has been inappropriate only define the ammount of on ml/kg and plateu.

Is more convenient to indicate at least the driving pressure or better the compliance oder a partition with esophageal baloon.

Patients often had multiple changes in their ventilator settings as their clinical status progressed and had evolving lung compliance over time. Given this we found it difficult to report specifics on ventilator or lung compliance in a manner that we thought would be instructive to providers. We do believe that our ventilator management across the system was consistent with current best practices and would be similar to standard of care ventilator management in similarly structured hospital systems. Our key message in writing the manuscript is to provide information on outcomes of mechanically ventilated patients rather than to attempt to provide guidance on specific ventilator management strategies. We have added a section to the limitations portion of the paper to highlight these issues. It reads “Finally, since patients frequently had evolving physiology with variable lung compliance and ventilator settings, we are unable to provide specific details on these parameters. However, our ventilator management was in keeping with current best practices and therefore we believe that our survival estimates would be reproducible in similar health care systems.” No patients had esophageal balloons placed given concerns for staff exposure to COVID-19. 

Which were the indication for ECMo? Murray score Oxigenation index....? And which was the ECMO capability in the number of the admission in ICU .

Our indications for ECMO initiation were in line with those recommended by ELSO. As stated in the text “Patients were considered for venovenous (VV) ECMO if age < 60 years old, on IMV < 10 days, had a P/F ratio < 100 and/or failed lung protective ventilation, despite neuromuscular blockade and prone positioning, or had recalcitrant hypercapnic acidosis affecting perfusion.” Like many other clinical decisions, ECMO was initiated only after a formal consultation by an experience ECMO intensivist and discussion with the cannulating cardiac surgeon. We were not reliant on a single parameter but did require all patients to fail conventional lung protective ventilation, neuromuscular blockade and proning. We expanded our typical ECMO capacity to the point that we were staffing up to 10 ECMO patients simultaneously. This is an increase from our typical maximum capacity of approximately 6 patients. 

Finally the big limitation you have also underline is the lack of long term survival which is important to evaluate the treatment.

Thank you for raising this important point. As mentioned above, we have updated outcomes for all patients included in the study. This makes our manuscript unique as it is one of the only studies providing complete outcomes data for critically ill COVID-19 patients.

---

## [Decision Letter · Decision Letter 1]

9 Nov 2020

Outcomes of Mechanically Ventilated Patients with COVID-19 Associated Respiratory Failure

PONE-D-20-21719R1

Dear Dr. King,

We’re pleased to inform you that your manuscript has been judged scientifically suitable for publication and will be formally accepted for publication once it meets all outstanding technical requirements.

Kind regards,

Chiara Lazzeri

Academic Editor

PLOS ONE

Additional Editor Comments (optional):

Reviewers' comments:

Reviewer's Responses to Questions

**Comments to the Author**

1. If the authors have adequately addressed your comments raised in a previous round of review and you feel that this manuscript is now acceptable for publication, you may indicate that here to bypass the “Comments to the Author” section, enter your conflict of interest statement in the “Confidential to Editor” section, and submit your "Accept" recommendation.

Reviewer #2: All comments have been addressed

2. Is the manuscript technically sound, and do the data support the conclusions?

Reviewer #2: Yes

3. Has the statistical analysis been performed appropriately and rigorously? 

Reviewer #2: Yes

4. Have the authors made all data underlying the findings in their manuscript fully available?

Reviewer #2: Yes

5. Is the manuscript presented in an intelligible fashion and written in standard English?

Reviewer #2: Yes

6. Review Comments to the Author

Reviewer #2: Thanks

Can you please put in the paper ECMO cannulation parameters you have indicated on the answer ?

7. PLOS authors have the option to publish the peer review history of their article (what does this mean?). If published, this will include your full peer review and any attached files.

Reviewer #2: No

---

## [Editor Report · Acceptance letter]

12 Nov 2020

PONE-D-20-21719R1 

Outcomes of Mechanically Ventilated Patients with COVID-19 Associated Respiratory Failure 

Dear Dr. King:

I'm pleased to inform you that your manuscript has been deemed suitable for publication in PLOS ONE. Congratulations! Your manuscript is now with our production department. 

Kind regards, 

on behalf of

Dr. Chiara Lazzeri 

Academic Editor

PLOS ONE